# You Only Evaluate Once:
# a Simple Baseline Algorithm for Offline RL

**Wonjoon Goo and Scott Niekum**
Department of Computer Science
University of Texas at Austin
{wonjoon,sniekum}@cs.utexas.edu

**Abstract:** The goal of offline reinforcement learning (RL) is to find an optimal policy given prerecorded trajectories. Many current approaches customize existing off-policy RL algorithms, especially actor-critic algorithms in which policy evaluation and improvement are iterated. However, the convergence of such approaches is not guaranteed due to the use of complex non-linear function approximation and an intertwined optimization process. By contrast, we propose a simple baseline algorithm for offline RL that only performs the policy evaluation step once so that the algorithm does not require complex stabilization schemes. Since the proposed algorithm is not likely to converge to an optimal policy, it is an appropriate baseline for actor-critic algorithms that ought to be outperformed if there is indeed value in iterative optimization in the offline setting. Surprisingly, we empirically find that the proposed algorithm exhibits competitive and sometimes even state-of-the-art performance in a subset of the D4RL offline RL benchmark. This result suggests that future work is needed to fully exploit the potential advantages of iterative optimization in order to justify the reduced stability of such methods.

**Keywords:** offline reinforcement learning, conservative policy evaluation

## 1 Introduction

The standard reinforcement learning setting involves an active component during learning: an agent continuously gathers experience as it learns. This is a very general learning framework that resembles the way animals learn, but the interactive component often hurts the applicability of RL since the agent interaction can be expensive or unsafe. To address this challenge, the offline reinforcement learning paradigm has been proposed, which aims to learn a policy purely from pre-generated data [1]. Considering that many recent breakthroughs in machine learning can be attributed to large-scale data, this new paradigm is very promising. However, the offline setup causes significant theoretic and algorithmic difficulties that need to be resolved to fulfill this promise.

Specifically, actor-critic based off-policy RL algorithms, which iterate policy evaluation and improvement, suffer from the overestimation problem caused by function approximation error and bootstrapping [2, 3] in the offline RL setup, even though the algorithms are equipped with algorithmic techniques [4, 2, 5] that can stabilize learning and mitigate the so-called *Deadly Triad* [6]. This is because over-estimated values cannot be readjusted in offline RL, unlike the ordinary RL setup where incorrectly optimistic actions get executed and corrected.

The Deadly Triad states that when off-policy learning, function approximation, and bootstrapping are used together, the danger of instability and divergence arises [6]. Therefore, the actor-critic algorithms that leverage all three components are vulnerable. Many algorithms try to stabilize learning by ensuring the queries for bootstrapping to be within the data manifold of the given offline dataset [7, 8, 9, 10, 11, 12] since over-estimation is prominent when the value function is queried for out-of-distribution inputs. However, strictly speaking, the Deadly Triad occurs due to iteration of the actor-critic algorithm; the first critic update is simply policy evaluation of the action-value function of the behavior policy $\beta$ using *on-policy* data. While the policy implied by the first value function $Q^\beta$ is likely to be suboptimal that would be surpassed by iterative algorithms, it has not been fully examined that a greedy policy with regard to $Q^\beta$ cannot work well in the offline RL setting.

5th Conference on Robot Learning (CoRL 2021), London, UK.

To this end, we propose a baseline algorithm, named YOEO, that evaluates the value of a behavior policy once and extracts a greedy policy under the learned value approximation, and by doing so, we examine the hidden presumption of the actor-critic based offline RL algorithms that the iterative process is essential and beneficial. Our algorithm leverages pessimism under uncertainty [13, 14] within the distributional RL framework [15, 16, 17]. Surprisingly, we find that the greedy policy extracted from the value function of the behavior policy with our algorithm exhibits competitive and sometimes even state-of-the-art results in D4RL benchmarks [18]. Our results indicate not only the effectiveness of the proposed algorithm adopting pessimistic regularization, but also implies potential vulnerabilities of the iterative optimization process of actor-critic algorithms in the offline setting, especially when a complex function approximator like a deep neural network is used. We expect that the proposed strong baseline will foster future offline RL research by allowing researchers to measure the actual advantage coming from the iterative process, which is a main concern in offline RL.

## 2   Related Work

Evaluating and improving a policy with the data generated from a different policy (off-policy RL) has been widely investigated, and several papers have shown theoretical convergence properties of prediction and control algorithms in the off-policy setting [19, 20, 21]. However, theoretical frameworks are limited to linear function approximation, while non-linear function approximation is essential to handle large-scale MDPs. Yet, there have been efforts to build practical algorithms with non-linear function approximators, and they have shown considerable success in various domains [22, 23, 24, 25] tackling real-world RL problems.

In theory, off-policy algorithms can be used in the offline setup without any modifications, but the algorithms often catastrophically fail when applied in the batch setting [7]. This is due to the accumulation of extrapolation error from bootstrapping and the policy improvement step (i.e. $\max$ operation) [7, 8]. In the non-batch setting, new experiences gathered via interaction can prevent this degenerate case, but it is impossible in the batch setting where interaction is prohibited. Pertaining to this problem, much research has been proposed, especially in the context of actor-critic algorithms in which policy evaluation and improvement are iterated. One class of solutions constrains the policy improvement step so that the optimized policy matches the behavior policy, in distribution [9] or in support [8]. In more recent work, a behavior policy for distribution matching is replaced with a prior policy, which is trained along with policy evaluation via weighted behavior cloning [12]. In [11], the policy improvement step is omitted while using the prior policy as a target policy for the evaluation. While the prior works try to overcome the Deadly Triad, we sidestep the problem and examine the hidden assumption of the actor-critic algorithm that the iterative process is essential and beneficial.

The most closely related prior works are behavior cloning-based methods [26, 10] which mimic a subset of good state-action pairs from the pre-generated trajectories. Since these methods approximate a value function for a behavior policy without an iterative policy improvement step, the algorithm does not diverge. The main difference with our proposed approach is that we train an action-value function with pessimistic regularization while the prior works evaluate a state- or action-value function to filter state-action pairs based on advantage calculated with Monte-Carlo return [26] or TD($k$) return [10] and train a policy via behavior cloning with the filtered data.

For principled regularization of the action-value function, we adopt pessimism under uncertainty, which can address the overestimation problem when the given offline dataset is not informative enough to estimate the value of every action given a state [14]. This pessimism can be applied by directly penalizing $Q$ values for a particular state-action distribution [13], or with model-approximation [27] that leverages uncertainty prediction techniques developed for supervised learning, such as Lakshminarayanan et al. [28] or van Amersfoort et al. [29]. While the previous approaches apply pessimism unconditionally for every state [13] or use transition-dynamics based proxies [27] for measuring uncertainty, we propose a theoretically justifiable regularization method for estimating $Q^\beta$ that is based on distributional RL.

## 3   Preliminaries

The common mathematical framework for reinforcement learning is a Markov Decision Process (MDP), which is defined by a tuple $\mathcal{M} = (\mathcal{S}, \mathcal{A}, T, d_0, r, \gamma)$ defined by a set of states $s \in \mathcal{S}$, a

set of actions $a \in \mathcal{A}$, conditional transition dynamics $T(s'|s,a)$, an initial state distribution $d_0$, a reward function $r : \mathcal{S} \times \mathcal{A} \to \mathbb{R}$, and a discount factor $\gamma \in (0,1]$. In this framework, the goal of reinforcement learning is to find an optimal policy $\pi(a|s)$ that maximizes an expected sum of discounted reward (return). Formally, the objective is defined as:

$$J(\pi) = \mathbb{E}_{\tau \sim p^\pi} \left[ \sum_{t=0}^{H} \gamma^t r(s_t, a_t) \right], \tag{1}$$

where $\tau$ is a sequence of states and actions $(s_0, a_0, \ldots, s_H, a_H)$ of length $H$, and $p^\pi$ is a trajectory distribution of a policy, which can be represented as:

$$p^\pi(\tau) = d_0(s_0) \prod_{t=0}^{H} \pi(a_t|s_t) T(s_{t+1}|s_t, a_t). \tag{2}$$

One way to find an optimal policy is to estimate an action-value function $Q^\pi$, which represents the expected return over possible trajectories following a policy $\pi$ starting from a given state and action: $Q^\pi(s_t, a_t) = \mathbb{E}_{\tau \sim p^\pi(s,a)} \left[ \sum_{t'=t}^{H} \gamma^{t'-t} r(s_{t'}, a_{t'}) \right]$. $Q^\pi$ function implies a greedy policy $\pi^+(a|s) = \delta(a = \arg\max_b Q^\pi(s,b))$, which is better than or equal to its original evaluation target policy $\pi$. Therefore, when we perform policy evaluation ($Q$ estimation) and policy updates iteratively, we can move toward the optimal policy $\pi^*$ and the optimal $Q$ function $Q^{\pi^*}$. Policy evaluation can be done with a Monte-Carlo method, but bootstrapping is commonly used, which utilizes a recursive equation that must be satisfied at convergence:

$$Q^\pi(s,a) = r(s,a) + \gamma \mathbb{E}_{s' \sim T(s'|s,a), a' \sim \pi(a'|s')} Q^\pi(s', a'). \tag{3}$$

When an MDP is discrete and $Q$ can be represented by a tabular representation (i.e. when $|\mathcal{S}| \times |\mathcal{A}|$ is small), it is known that policy evaluation converges to a correct solution in the limit of the number of transition tuples [6]. However, when an MDP has a large state or action space, $Q$ has to be represented with a function approximator, such as a deep neural network. In addition, when the action space is continuous, directly extracting a better policy from $Q$ becomes infeasible due to the $\arg\max$ operator. These restrictions are addressed in actor-critic algorithms [4, 2, 5] which explicitly alternate policy evaluation and policy improvement with a batch of (online) transition samples $D$ and a parameterized value function $\hat{Q}_\theta$ and a policy $\pi_\phi$:

$$\theta^{k+1} \leftarrow \arg\min_\theta \mathbb{E}_{s,a,s' \sim D} \left[ d\big(\hat{Q}_\theta(s,a), r(s,a) + \gamma \mathbb{E}_{a' \sim \pi_{\phi^k}(a'|s')} \hat{Q}_{\theta^k}(s', a')\big) \right] \text{ (policy evaluation)}, \tag{4}$$

$$\phi^{k+1} \leftarrow \arg\max_\phi \mathbb{E}_{s \sim D, a \sim \pi_\phi(a|s)} [\hat{Q}_{\theta^{k+1}}(s,a)] \text{ (policy improvement)}, \tag{5}$$

where $k$ is an update step, and $d$ is a distance measure such as squared $l_2$ or Huber loss.

## 4  You Only Evaluate Once

Behind actor-critic based offline RL algorithms, there is a common presumption that the iterative process is essential in achieving better performance than that of the behavior policy, even though it could sacrifice the reliability of the algorithm due to the over-estimation problem. This is because we want to find a policy that behaves differently from the data-generating policy by making counterfactual queries (policy improvement) and answering (policy evaluation) them iteratively [1]. However, it has not been established that a simpler and safer baseline cannot work well—a policy that selects the best action with regard to the action-value function of the *behavior policy*. Without rigorously examining this hypothesis, the true worth of iterative algorithms and counterfactual queries cannot be fully understood.

In this paper, we challenge the iterative offline algorithms by proposing a stable offline algorithm that recovers the best action of behavioral policy $\beta(s)$ that is used to generate an offline dataset $D$. Formally, our goal is to find the greedy *behavioral* policy $\beta^*(s)$ that selects the best action with respect to $Q^\beta$, only considering the action candidates of $\beta$:

$$\beta^*(s) := \arg\max_{a \in B(s)} Q^\beta(s,a) \quad \text{where} \quad B(s) = \operatorname{supp} \beta(s) := \{a \in \mathcal{A} : \beta(a|s) > 0\}. \tag{6}$$

When we have an oracle $B(s)$, $\beta^*$ can be estimated by simply making on-policy queries to the approximated action-value function $\hat{Q}^\beta$, which can be trained via TD loss based on SARSA tuple $(s, a, r, s', a') \in D$. This can be reasonably correct as long as the given dataset is sufficient to perform an on-policy evaluation. However, the approach to learning $\beta^*$ directly by estimating $B(s)$ and $\hat{Q}^\beta$ is prone to failure. This is because directly modeling $B(s)$ is often infeasible since $\beta$ can be a mixture of many stochastic policies, and the estimation error in $B(s)$ can create unpredictable errors in finding $\beta^*$ because the prediction of $\hat{Q}^\beta$ for off-policy input $(s, \tilde{a})$ solely depends on the generalization ability of $\hat{Q}^\beta$. Figure 1 demonstrates one failure case in estimating $Q^\beta$; when $\hat{Q}^\beta$ is trained without any regularization, it behaves more like a state value function $V^\beta(s)$ ignoring the action input. Therefore, any erroneous action generated by estimated $B(s)$ will be treated as the best action.

Instead, we propose to learn the greedy behavioral policy $\beta^*$ without directly estimating $B(s)$. It can be achieved when $\hat{Q}^\beta$ is properly regularized so that the greedy policy implied by the regularized $\hat{Q}^\beta$ is $\beta^*$. We argue that *pessimistic* regularization satisfies such property:

**Theorem** ($\hat{Q}^\beta$ implies $\beta^*$)**.** *Let* $\hat{Q}^\beta$ *be a **valid pessimistic approximation** of* $Q^\beta$*: (1)* $\hat{Q}^\beta(s, \tilde{a}) < \max_{a \in B(s)} Q^\beta(s, a) \ \forall\, s \in D, \ \tilde{a} \notin B(s)$ *and (2)* $\hat{Q}^\beta(s, a) = Q^\beta(s, a) \ \forall\, (s, a) \in D$. *Then, a greedy policy over* $\hat{Q}^\beta$ *is* $\beta^*$.
*Proof.*
$$\arg\max_{a \in \mathcal{A}} \hat{Q}^\beta(s, a) \;=\; \arg\max_{a \in B(s)} \hat{Q}^\beta(s, a) \;=\; \arg\max_{a \in B(s)} Q^\beta(s, a) \;=\; \beta^*(s). \qquad \square$$

Pessimistic regularization is essential even for learning $Q^\beta$ since there are always out-of-distribution state-action pairs $(s, \tilde{a})$ given a fixed dataset $D$, and the values of these inputs depend on the generalization characteristic of the function approximator used to learn $Q^\beta$. Therefore, we have to regularize $\hat{Q}^\beta$ properly, and we adapt pessimism under the uncertainty principle since we cannot assume any value for OOD input unless we have a prior regarding an MDP. Yet, in contrast to pessimistic regularization for $Q^*$, which requires a calibrated epistemic uncertainty measurement [14] or suffers from over-regularization due to excessive pessimism [13], $\hat{Q}^\beta$ can be properly regularized by enforcing $\hat{Q}^\beta(s, \tilde{a})$ to be smaller than the maximum value of $\hat{Q}^\beta(s, a)$ for $a \in B(s)$.

### 4.1 Practical Implementation

When we have a valid pessimistic approximation of $Q^\beta$, $\beta^*$ can be inferred without knowing $B(s)$ since the greedy action over the approximated value function $\hat{Q}^\beta$ is $\beta^*$. However, the first constraint of the valid pessimistic approximation still depends on $B(s)$, and it makes implementing the pessimistic regularization in its original form difficult.

To address this problem, we express the two conditions of the theorem in terms of the state value *distribution* over $Y^\beta(s)$; $Y^\beta(s)$ is a random variable representing the return of the policy $\beta$ starting from a state $s$ where its expectation is the state value $V^\beta(s) = \mathbb{E}[Y^\beta(s)]$ [15, 16, 17]. With the random variable $Y^\beta(s)$, the two constraints can be rewritten as follows: (1) $\hat{Q}^\beta(s, \cdot) \leq \max_{a \in B(s)} Q^\beta(s, a) = \max_w Y^\beta(s; w)$, and (2) $\hat{Q}^\beta(s, a) = r + \gamma \mathbb{E}_{s' \sim T(s, a)}[V^\beta(s')] = r + \gamma \mathbb{E}_{s' \sim T(s, a)}[\mathbb{E}_w[Y^\beta(s'; w)]]$ where $w$ is an outcome of the sample space of $Y$. This converted expression is defined without a need for $B(s)$, and therefore, we can now implement the pessimistic regularization with $Y^\beta(s)$ instead of $B(s)$. Note that the constraint is defined with the maximum and the expected value of $Y^\beta$ only for the state $s \in D$. This allows us to use any on-policy evaluation methods for $Y^\beta$ as long as the resulting approximation is precise for on-policy states.

We propose a two-step approach: we first perform state value *distribution* learning using $(s, r, s')$ transition tuple in $D$ (behavioral policy evaluation) and then train $\hat{Q}^\beta$ in a supervised manner that satisfies the given constraints. Specifically, we represent a state-value distribution of $Y^\beta$ using implicit quantile network (IQN) [17], which is a recently proposed distributional RL algorithm that models the value distribution in the form of an inverse cumulative distribution function parameterized with $\psi$: (with a slight abuse of notation) $\hat{Y}_\psi(s; \tau) = \inf\{v \in \mathbb{R} : \tau \leq \mathrm{Prob}(Y(s) \leq v)\}$ where $\tau$ is queried probability. In IQN, the parameter $\psi$ can be trained by a distributional TD learning. For details about the distributional TD learning, please refer to Will et al. [17].

With the approximated state-value distribution $\hat{Y}_\psi$ of the behavior policy $\beta$, we train an action-value function $\hat{Q}_\theta$ with a supervised loss for on-policy state-action pair $(s, a)$ and a pessimistic regularization $\mathcal{R}$ for off-policy pair $(s, \tilde{a})$:

$$L = \mathbb{E}_{(s,a,r,s')\sim D}\left[\frac{1}{2}\left|\hat{Q}_\theta(s,a) - \left(r + \gamma\mathbb{E}[\hat{Y}_\psi(s')]\right)\right|^2 + \lambda\mathcal{R}(s;\theta)\right] \tag{7}$$

where $\lambda$ is a hyperparameter that controls the strength of pessimism. While we can approximate the expectation of $\hat{Y}_\psi$ with sampling, we use the median value of the distribution $\hat{Y}_\psi(s'; 0.5)$ for computational efficiency.

The main goal of $\mathcal{R}$ is to implement the pessimism on $\hat{Q}_\theta$ and guarantee $\hat{Q}_\theta$ to be a valid pessimistic approximation of $Q^\beta$ that satisfies $\hat{Q}_\theta(s, \cdot) \leq \max Y^\beta(s)$. While every possible action $a \in \mathcal{A}$ needs to be checked, it is computationally infeasible when the action space is large or infinite. Hence, we sample actions from a static distribution $\mu$ (passive pessimistic regularization) and a distribution $\pi_\phi$ that is actively changing over training iterations (active pessimistic regularization). Then, we penalize $\hat{Q}_\theta$ if the estimations on the action samples violate the pessimistic constraint. We use a different hyperparameter $\tau$ for each of action samples. We sample $n_b$ number of actions from each distribution and use the log-sum-exp trick to change the hard constraint into a soft constraint:

$$\mathcal{R}(s;\theta) = \log\left[\exp(\hat{Y}_\psi(s;\tau_1)) + \sum_{\tilde{a}\sim\pi_\phi(s)}\exp\hat{Q}_\theta(s,\tilde{a})\right] + \log\left[\exp(\hat{Y}_\psi(s;\tau_2)) + \sum_{b\sim\mu(s)}\exp\hat{Q}_\theta(s,b)\right]. \tag{8}$$

We use a random uniform policy for $\mu$ while the $\pi_\phi$ is trained against trained $Q$ (or a set of $Q$s when an ensemble technique is used) as same as an ordinary actor in the actor-critic algorithm. However, it is different from the actor in that $\pi_\phi$ works as an active regularizer that adversarially finds the wrong generalization part of $\hat{Q}_\theta$. For $\tau_1$ and $\tau_2$, we use 0.9 and 0.1 respectively. While the $\tau_1$ is decided following the definition of valid pessimism with a safe margin of 0.1, we use a small $\tau_2$ since we expect that a random policy will perform poorly. We train multiple $\hat{Q}_\theta$ models and aggregate them by taking a min over

---

**Algorithm 1:** You Only Evaluate Once

**Input :** Dataset $D = \{(s, a, r, s')\}$,
         Hyper-parameter $\lambda, \tau_1, \tau_2$
Initialize $\psi, \theta$, and $\phi$
Train $\hat{Y}_\psi$ with distributional TD-loss
**while** *until convergence* **do**
    Update $\theta$ with $\nabla_\theta L$ (Eq. 7)
    Update $\phi$ with $\nabla_\phi\mathbb{E}_{s\sim D}\hat{Q}_\theta(s, \pi_\phi(s))$
**end**
**return** $\hat{Q}_\theta, \pi_\phi, \tilde{\pi}$

---

inferred $\hat{Q}_\theta$ values to get a robust value estimate. We train multiple models with different initial parameters and the same data following Osband et al. [30].

As an approximation of $\beta^*$, we use two policies defined on top of $\hat{Q}_\theta$. One is the last $\pi_\phi$ we have at the end of training since $\pi_\phi$ is directly trained to find the greedy policy over $\hat{Q}_\theta$. If $\hat{Q}_\theta$ is properly regularized (i.e. $\hat{Q}_\theta$ is a valid pessimistic approximation), $\pi_\phi$ will converge to $\beta^*$. Therefore, the policy will be implicitly constrained to use actions limited to the support of the behavior policy $\beta$. The other policy is a nonparametric policy $\tilde{\pi}$ whose action space is explicitly constrained to the actions shown in the dataset $D$: $\tilde{\pi}(s) = \arg\max_{a\sim D}\hat{Q}_\theta(s,a)$. $\tilde{\pi}$ can perform better than $\pi_\phi$ when the pessimistic constraint is hard to achieve due to restriction of a dataset or characteristics of an MDP, such as the dataset size $|D|$ or the large action space $|\mathcal{A}|$, that allows room for adversarial attacks; when there is much room for adversarial attacks, $\pi_\phi$ will continuously try to regularize $\hat{Q}_\theta$ by suggesting diverse adversarial actions while it does not converge to the desired policy $\beta^*$. In such a case, directly utilizing $\pi_\phi$ would result in poor performance, but the action set restricted policy $\tilde{\pi}$ can perform well by testing limited actions that are more likely to belong to $B(s)$. The overall algorithm is shown in Algorithm 1, and the implementation details are provided in Appendix.

Since the training of $\hat{Q}_\theta$ largely depends on the trained state value distribution $\hat{Y}_\psi$, the correctness of $\hat{Y}_\psi$ is essential in learning greedy behavioral policy $\beta^*$. Fortunately, since we are performing policy evaluation with on-policy data, we avoid the Deadly Triad, and it is more likely to converge to a correct $Y^\beta$ than other methods that perform off-policy learning. Furthermore, the correctness of $\hat{Y}_\psi$ and the regularized $\hat{Q}_\theta$ can be roughly tested by comparing the estimated value with the Monte-Carlo

return. This is an extra debugging feature that is only available in YOEO, and we can leverage this to set the hyperparameters.

It is noteworthy that we are leveraging aleatoric uncertainty to implement pessimism. When epistemic uncertainty of action-value function $Q^*(s, a)$ can be estimated with a fixed dataset, using it to penalize the value function is a theoretically justifiable implementation of pessimism [14]. However, measuring epistemic uncertainty is still an open problem when a deep neural network is used even in the simpler supervised learning setting that does not include bootstrapping as RL. For that reason, pessimism is commonly implemented with a proximal objective, such as learning a lower bound of the true $Q^*$ [13] or using an uncertainty proxy [27, 31], and therefore, these approaches often require sensitive hyperparameter tuning to find the right level of pessimism [1]. In contrast, the objective of YOEO allows us to use aleatoric uncertainty that can be directly estimated. Also, the two-step training prevents the error from propagating through bootstrapping, so our method is less susceptible to divergence at the cost of optimality. This makes our algorithm a suitable *baseline* for offline RL that achieves stability at the cost of optimality.

## 5    Experiments

We aim to study the following question: how much performance gain do potentially risky policy iteration algorithms provide compared to more stable baseline algorithm YOEO? We compare the performance of YOEO against several baselines and prior works based on policy iteration: behavior cloning (BC), soft actor-critic (SAC) [5] without interaction, bootstrapping error accumulation reduction (BEAR) [8], behavior regularized actor-critic (BRAC) [9], advantage weighted regression (AWR) [10], batch constrained deep Q-learning (BCQ) [7], and conservative Q-learning (CQL) [13].

The comparison is made on a subset of datasets in the D4RL offline RL benchmark [18]. We use MuJoCo [32] locomotion tasks [33], Adroit hand-manipulation tasks [34], and Franka kitchen [35] tasks. For MuJoCo locomotion tasks, we use three environments (`hopper`, `walker2d`, and `halfcheetah`) in four different settings (`random`, `medium-replay`, `medium`, and `medium-expert`). For Adroit hand-manipulation tasks, we use four environments (`pen`, `door`, `relocate`, and `hammer`) in two settings (`human` and `cloned`). For Franka kitchen tasks, we use three different settings (`complete`, `partial`, and `mixed`). We train YOEO for 1 million stochastic gradient steps for both $\hat{Y}_\psi$ and $\hat{Q}_\theta$, then we report the average normalized performance score over 100 trajectories. The results are displayed in Table 1.

Despite the fact that YOEO aims to learn a greedy behavioral policy $\beta^*(s)$ with respect to $Q^\beta$ rather than try to learn $\pi^*$, it shows highly competitive results: it achieves better performance than the current state-of-the-art model-free offline RL algorithms in five datasets (hopper-random, walker2d-medium-replay, hopper-medium-expert, and hammer-cloned, kitchen-compete), and competitive performance across all other configurations surpassing most of the other offline RL algorithms. The results indicate that the previous offline RL algorithms fail to fully exploit the potential benefit of iterative evaluation and update.

The performance of $\tilde{\pi}$ shown in the last column of Table 1 also supports our hypothesis on $\pi_\phi$. While $\tilde{\pi}$ shows similar results in other datasets, $\tilde{\pi}$ shows a better result than $\pi_\phi$ in -human type dataset in Adroit tasks and -parital and -mixed type datasets in the kitchen task of which the size of the dataset is relatively small or the action space is large; since $\hat{Q}_\theta$ needs to be regularized for diverse set of actions, $\pi_\phi$ would not converge to $\beta^*$ whose actions are limited to the support of $\beta$.

### 5.1    Ablation Study: how much will each regularization term affect the performance?

We conduct an ablation study to show the effectiveness of the regularization and the contribution of each term in Eq. 8. Specifically, we consider two ablations: (1) $\hat{Q}^\beta$ trained only via TD loss based on $(s, a, r, s', a')$ tuple without any regularization (denoted as $\hat{Q}^\beta \backslash \mathcal{R}$) and (2) $\hat{Q}^\beta$ trained without the second term in Eq. 8 (denoted as $\hat{Q}^\beta \backslash \mu$). Additionally, we test the effect of the ensemble by changing the number of trained models. The experiment settings and the results are summarized in Table 2.

We confirm the necessity of pessimistic regularization even when we do on-policy policy evaluation; the qualitative results shown in Figure 1 reveal that $\hat{Q}_\theta \backslash \mathcal{R}$ can estimate the ground-truth value for on-policy input $(s, a)$, but it fails to estimate the value for out-of-distribution input $(s, b)$, especially

Table 1: Performance of YOEO and prior methods on a subset of D4RL benchmarks. Each number represents the performance relative to a random policy as 0 and an expert policy as 100. All the numbers except ours are borrowed from [18] and [13]. The numbers of our results are averaged over 3 different random seeds.

| Type | Environemt | BC | SAC-offline | BEAR | BRAC | AWR | BCQ | CQL $(\mathcal{H})$ | YOEO $(\pi_\phi)$ | YOEO $(\bar{\pi})$ |
|------|-----------|----|----|----|----|----|----|----|----|----|
| Random | HalfCheetah | 2.1 | 30.5 | 25.1 | 31.2 | 2.5 | 2.2 | 35.4 | 4.5 | 5.7 |
| | Hopper | 9.8 | 11.3 | 11.4 | **12.2** | 10.2 | 10.6 | 10.8 | **12.2** | 12.3 |
| | Walker2D | 1.6 | 4.1 | **7.3** | 1.9 | 1.5 | 4.9 | 7 | 4.9 | 4.1 |
| Medium-Replay | HalfCheetah | 38.4 | -2.4 | 38.6 | **47.7** | 40.3 | 38.2 | 46.2 | 36.2 | 30.9 |
| | Hopper | 11.8 | 3.5 | 33.7 | 0.6 | 28.4 | 33.1 | **48.6** | 42.5 | 37.2 |
| | Walker2D | 11.3 | 1.9 | 19.2 | 0.9 | 15.5 | 15 | 32.6 | **41.9** | 30.1 |
| Medium | HalfCheetah | 36.1 | -4.3 | 41.7 | **46.3** | 37.4 | 40.7 | 44.4 | 45.1 | 43.6 |
| | Hopper | 29 | 0.8 | 52.1 | 31.1 | 35.9 | 54.5 | **86.6** | 71.9 | 62.5 |
| | Walker2D | 6.6 | 0.9 | 59.1 | **81.1** | 17.4 | 53.1 | 74.5 | 74.1 | 76.2 |
| Medium-Expert | HalfCheetah | 35.8 | 1.8 | 53.4 | 44.2 | 52.7 | **64.7** | 62.4 | 41.9 | 35.9 |
| | Hopper | 111.9 | 1.6 | 96.3 | 0.8 | 27.1 | 110.9 | 111 | **112.1** | 112 |
| | Walker2D | 6.4 | -0.1 | 40.1 | 81.6 | 53.8 | 57.5 | **98.7** | 89.5 | 84.3 |
| human | pen | 34.4 | 6.3 | -1 | 8.1 | 12.3 | **68.9** | 37.5 | 17.3 | 43.6 |
| | door | 0.5 | 3.9 | -0.3 | -0.3 | 0.4 | 0 | **9.9** | -0.1 | 5.7 |
| | relocate | 0 | 0 | -0.3 | -0.3 | 0 | -0.1 | 0.2 | -0.1 | 0.2 |
| | hammer | 1.5 | 0.5 | 0.3 | 0.3 | 1.2 | 0.5 | 4.4 | 2.6 | **5.4** |
| cloned | pen | **56.9** | 23.5 | 26.5 | 1.6 | 28 | 44 | 39.2 | 32.6 | 30.5 |
| | door | -0.1 | 0 | -0.1 | -0.1 | 0 | 0 | 0.4 | 0.1 | **0.7** |
| | relocate | -0.1 | -0.2 | -0.3 | -0.3 | -0.2 | -0.3 | -0.1 | -0.1 | -0.2 |
| | hammer | 0.8 | 0.2 | 0.3 | 0.3 | 0.4 | 0.4 | 2.1 | **2.7** | 1.2 |
| kitchen | complete | 33.8 | 15.0 | 0 | 0 | 0 | 8.1 | 43.8 | **63.2** | 31.3 |
| | partial | 33.8 | 0 | 13.1 | 0 | 15.4 | 18.9 | **49.8** | 17.2 | 46.8 |
| | mixed | 47.5 | 2.5 | 47.2 | 0 | 10.6 | 8.1 | **51.0** | 5.9 | 40.4 |

Table 2: Ablation experiment results. The normalized performance over 3 random seeds is displayed.

| | | $\hat{Q}_\theta \backslash \mathcal{R}$ | $\hat{Q}_\theta \backslash \mu(s)$ | $\hat{Q}_\theta$ **(Ens. 1)** | $\hat{Q}_\theta$ **(Ens. 3)** | YOEO |
|--|--|--|--|--|--|--|
| Supervised w/ $\hat{Y}_\psi$ | | ✗ | ✓ | ✓ | ✓ | ✓ |
| $\mathcal{R}$ w/ $\mu(s)$ | | ✗ | ✓ | ✓ | ✓ | ✓ |
| $\mathcal{R}$ w/ $\pi(s)$ | | ✗ | ✗ | ✓ | ✓ | ✓ |
| # Ensembles | | 5 | 5 | 1 | 3 | 5 |
| Medium-Replay | Hopper | 30.6 | **49.1** | 21.3 | 36.5 | 42.5 |
| | Walker2D | 3.7 | 21.2 | 18.9 | 28.2 | **41.9** |
| Medium | Hopper | 2.7 | 14.9 | 54.6 | 65.9 | **71.9** |
| | Walker2D | -0.2 | 29 | 66.3 | 75.9 | 74.1 |
| Medium-Expert | Hopper | 10.4 | 58.2 | 89.6 | 112 | **112.1** |
| | Walker2D | 3 | 0.3 | 62.2 | **94.9** | 89.5 |

when the given dataset is homogeneous such as the `-medium` dataset whose behavior policy $\beta$ is not a mixture of different policies. This is because $\hat{Q}^\beta$ behaves more like a $\hat{V}^\beta$, ignoring the action, since an effective action $a$ is predictable based on $s$. The performance degradation of $\hat{Q}_\theta \backslash \mu$ indicates that a random policy $\mu$ is a good heuristic that can prevent the degeneration of the action-value function; the second regularization term using $\mu$ can foster the discrimination ability of the value function by enforcing $\hat{Q}_\theta$ to estimate a different value for $(s, a)$ and $(s, b)$, and this can increase the performance significantly especially when the given dataset is generated with a homogeneous policy.

We also observe the performance benefit of training more models and ensembling them, especially for `-medium-replay` type datasets. We hypothesize that YOEO's pessimistic regularization method has high variance due to the stochasticity of the loss function, and the ensembling technique can improve learning by enabling robust prediction for off-policy data from which the stochasticity is derived.

## 5.2 Why do CQL and other methods fail?

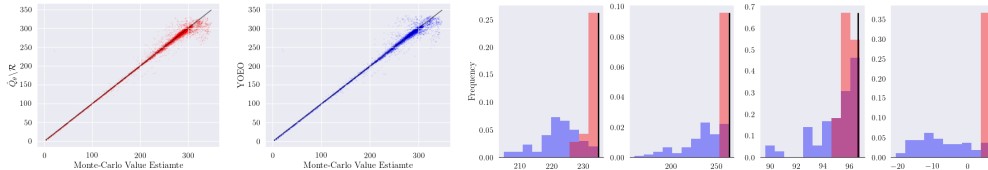

Figure 1: Qualitative evaluation of $\hat{Q}_\theta \backslash \mathcal{R}$ (red) and $\hat{Q}_\theta$ (YOEO, blue) trained for `hopper-medium-v0` dataset. In the left two figures, we plot the predictions of each method for 10,000 on-policy data points $(s, a)$. The x-axis is the Monte-Carlo value estimate of the given data point. In the right, we plot the predictions of $(s, \mu(s))$ for 4 randomly sampled states from $D$. 100 actions are sampled from random uniform policy $\mu$. Monte-Carlo estimate of the given state is drawn with the black line, and a histogram of the predictions with each method is drawn with the corresponding colors. While both $\hat{Q}_\theta \backslash \mathcal{R}$ and $\hat{Q}_\theta$ estimate the value precisely for the on-policy input, $\hat{Q}_\theta \backslash \mathcal{R}$ behaves more like $V^\beta(s)$ failing to take the action input into account for a value prediction.

YOEO outperforms the current state-of-the-art methods, specifically CQL, in several datasets. Considering that the pessimistic regularization of YOEO is milder than CQL, we hypothesize that the failure of CQL to surpass $\beta^*$ is derived from over-regularization; YOEO only penalizes $\hat{Q}_\theta(s, b)$ that are larger than the maximum value of the state value distribution $\max \hat{Y}_\psi$, while CQL performs unbounded minimization for $\hat{Q}^*(s, b)$ and maximization for $\hat{Q}^*(s, a)$. When $b$ is sampled from a policy based on $\hat{Q}$, the regularization will practically penalize $\hat{Q}^*(s, b)$ until the value becomes smaller than $\hat{Q}^*(s, a)$. Even when $\hat{Q}^*(s, a)$ is smaller than $\max \hat{Y}_\psi$, CQL

Table 3: Experimental results of CQ$^\beta$L and its variants. MR, M, and ME represent medium-replay, medium, and medium-expert dataset types, and H and W represent Hopper and Walker2D environments respectively. The averaged normalized performance over 3 random seed is displayed.

|    |   | YOEO | CQL($\rho$) | CQ$^\beta$L($\rho$) | MOReL |
|----|---|------|------|------|------|
| MR | H | 42.5 | 26.9 | 29.7 | **93.6** |
|    | W | 41.9 | 13.3 | 26.7 | **49.8** |
| M  | H | 71.9 | 31.7 | 32 | **95.4** |
|    | W | 74.1 | **78.8** | 65 | 77.8 |
| ME | H | **112.1** | 111.9 | 111.5 | 108.7 |
|    | W | 89.5 | 63 | 83 | **95.6** |

would still induce more pessimism than YOEO. Furthermore, when $\hat{Q}^*$ is over-regularized, we can expect that $\hat{Q}^*$ would behave more like $\hat{Q}^\beta$, losing the advantage of the iterative process.

To confirm this hypothesis, we modify CQL to estimate $Q^\beta$ instead of $Q^*$ by changing the TD loss to be computed with $(s, a, r, s', a')$, not $(s, a, r, s')$. The pessimistic regularization part remains intact. We denote this algorithm CQ$^\beta$L. We ran the CQL($\rho$) algorithm, and the results are displayed in Table 3. CQ$^\beta$L($\rho$) works similarly to CQL($\rho$), and this supports our hypothesis that the CQL is over-regularizing. The success of the model-based offline algorithm (MOReL [31]), implies that the over-regularization of CQL can be avoided by utilizing the uncertainty that CQL lacks.

## 6 Discussion

We investigate a simple baseline algorithm for offline RL that only evaluates the value function of the behavior policy as opposed to approximate $Q^*$ with unstable iterative process. Since the proposed algorithm does not involve a policy optimization step and value re-evaluation based on the updated policy, the algorithm can be stable, but the resulting policy is more likely to be suboptimal. This makes the algorithm an appropriate baseline for actor-critic algorithms that ought to outperform this baseline if there is indeed value in iterative optimization in the offline setting. In the experiments, the proposed baseline surprisingly shows competitive results on the several D4RL benchmarks, surpassing the state-of-the-art results in some tasks. This implies the usefulness of conservativeness under uncertainty, which can prevent incorrect generalization behavior of a complex function approximator occurring due to lack of data, as well as the potential flaws of iterative optimization in actor-critic algorithms in the offline setting. Therefore, it is essential for future work to build a theoretical framework that sheds light on iterative optimization and generalization of offline actor-critic methods that use deep neural networks, if iterative optimization is to be fully taken advantage of.

**Acknowledgments**

This work has taken place in the Personal Autonomous Robotics Lab (PeARL) at The University of Texas at Austin. PeARL research is supported in part by the NSF (IIS-1724157, IIS-1638107, IIS-1749204, IIS-1925082), ONR (N00014-18-2243), AFOSR (FA9550-20-1-0077), and ARO (78372-CS).

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
