# OpenReview forum: "You Only Evaluate Once: a Simple Baseline Algorithm for Offline RL"
_robot-learning.org/CoRL/2021/Conference — CoRL2021 Poster_

### Official Review · Reviewer_q44u · 2021-07-23

**Originality:** Good
**Technical Quality:** Good
**Clarity Of Presentation:** Good
**Impact:** 3

**Recommendation:**

Strong Accept: I recommend accepting the paper and will argue for my recommendation even if other reviewers hold a different opinion.

**Summary:**

This paper proposes an offline RL method thas is supposed to be the well-designed baseline for other offline policy-learning algorithms.  It's goal is to effectively recover an optimal policy relative to the value function describing the behavior policy, that is to /not/ perform any explicit policy improvement.  The reasoning is that it is primarily the policy improvement step that induces instability by aggressively sampling out-of-distribution updates.

The general method is to learn a state-value function Y from sarsa updates, and then to learn a Q-funciton with both on-policy pairs and off-policy pairs, with the off-policy pairs having an extra regularization function.

A policy is then found using this Q-function (not sure how).

**Issues:**

l.29 Please define the deadly triad.  I know it's a thing, I had to look it up to get the exact definition.
Eq 6. I found this equation a bit confusing.  If \Beta(s) is your behavior policy that you are learning, I would have much preferred a \pi notation.  I am still a bit confused which is the learnt policy and which is the demonstrated policy.  Also please define supp someplace, it's not a super common notation (I'm assuming it's the set of possible actions that the behavior policy has taken in state s).
l145 in the stated theorem, shouldn't Q^hat be strictly less than the max? Otherwise Q-values for \tilde{a} could be equal to the max, and the argmax would but undefined (or multiple), which would not guarantee the reproduction of \Beta^*.  Perhaps having a margin would guarantee that the non-expert values stay below the max.
l.160 Could you consider using V vor your state value distribution? It took me a good 5 minutes to understand what you meant it to be, it could very well be called V^\Beta, this would greatly increase legibility.
l.189 Can you clarify Y(s; t_{1,2})'s meaning? Does a smaller tao push towards a less probable distribution? I should not have to read the IQN paper to understand what you are doing here ideally.


**Reviewer Expertise:**

Good: General knowledge of the area

**Strengths And Weaknesses:**

I really enjoy this paper's idea, and its questioning of many offline RL algorithms.   I think it should be accepted, and the results are important for the community.

....however, I found it often confusing to read, there are a lot of things going on and it's not always clear /why/.  For example, the naive reader would assume that running Q-learning SARSA would be sufficient to have a stable Q-function that one could then find an optimal policy for either through CEM (such as QT-opt), or by learning a stochastic BC agent and then using that as the support of \Beta(s) (this might actually reduce back to Critic-Reguralized Regression, but I've run out of time to check).

Instead of doing this you decide to go the pessimism route, which is one of the criticisms later on of similar offline PI algorithms.  This was a bit surprising turn of events that I think could be more justified.  I also am not clear why you then choose to use a <= instead of < or even <-epsilon to enforce your pessimistic constraint.  Clarification here would be welcome.

Eq. 8 is basically the core of the paper, IIUC the \lambda is zero if the pair is on-policy and some value >0 if it is off-policy.  If this is true it sholud be explicit in Eq. 8 (I'm not currently sure it's true).

As for R I clearly don't understand how it works, it includes \ph_\phi and \u(s), I don't fully understand how these are produced.  Are you using direct gradient updates from the critic to learn your actor?  Please make this more clear.  I also think the Y(s,tao) notation needs to be clarified,  it probably only takes one sentence but would make the reader's life much easier.  Additionally, I mention it below, but pleas call this function V.  I have to rewrite Y->V in my head for things to make sense and it took me a while to understand things the first pass because of this (FYI, this was the paper that took me the longest to review and I had to come back to it because it was hard to parse).

Once all this comes together, it would be nice to clearly explain how you actuall produce the final policy, ideall ywith a clear equation.  When reading the paper for the first pass, I'm clearly looking for "\pi = ________" to understand what's going on.  Explicit is your friend :)

Overall the experiments look reasonable, but I was a bit dissapointed to see that the figure I was primarily looking for (Performance of Q^B\R) is i the appendix.  This figure is quite important because it justifies why I need to put it with all this regularization stuff and not just run SARSA, I would suggest trying to fit it in if possible.

**Summary Of Recommendation:**

I put my recommendations in strength & weaknesses.  My main suggestion would be to rewrite Sec 4., especially Sec 4.1, and don't move so quickly, justify your choices, clean up the notation, explain a couple things more explicitly and say less on the unimportant things.  I don't know what the 1-wasserstein metric is, and I don't really care in this context, but because you wrote that you're doing 1-wasserstein metrics now my head is thinking about OT and I'm wondering if I totally missed something in the paper and then I'm lost.  I would replace lines 171-173 with something much more important: what in the world does the second parameter of your value function mean and do!?  Now that would have made me a happier and less confused reader :)

Overall I will gladly move to a strong accept, but I think it's important to clear up Sec 4 and 4.1 so that this is a solid and clear paper.

---

> ### Author Response · Authors · 2021-08-22
> **To Reviewer q44u [3/3]**
>
> ### How does the regularization work with the state value distribution $Y(s;\tau)$; What are $\mu$ and $\pi_\phi$, and how are these found?; What is the final policy?
>
> The regularization term $\mathcal{R}$ is to enforce the constraint of the theorem $\hat{Q}^\beta(s,a') \leq \max_{a \in B(s)} Q^\beta(s,a)$, and it is done by expressing the right-hand side of the inequality with $Y^\beta$: $\hat{Q}^\beta(s,a') \leq Y^\beta(s;1-\epsilon)$. Now, the problems are (1) how to enforce the constraint and (2) which action $a' \in \mathcal{A}$ to check since $\mathcal{A}$ can be very large.
>
> For the first problem, we choose to use the log-sum-exp trick, which converts the hard constraint into the soft constraint (Eq. (9)); when the soft maximum of $\hat{Q}(s,\cdot)$ over a set of actions $\{a'\}$ is larger than $Y^\beta(s;1-\epsilon)$, the soft maximum will be minimized.
>
> For the second problem, we use the actor $\pi_\phi$ trained with the current $\hat{Q}_\theta$ as an ordinary actor-critic algorithm (See Algorithm 1 in L197-198 for the equation). This is an effective choice since $\pi_\phi$ is an adversarial network that actively finds the $a'$ which potentially violates the constraint. This consists of the first term in Eq. (9). We use $\epsilon=0.1$ as a safety margin, i.e. $\tau_1=0.9$. When $\hat{Q}_\theta$ is properly regularized, $\pi_\phi$ converges to $\beta^*$ as the theorem states. Therefore, we report the performance of the last $\pi_\phi$ we get at the end of training.
>
> On top of that, we additionally regularize Q with a random uniform action distribution $\mu$ to ensure the constraint is met. The motivation behind using the static action distribution $\mu$ is that we can enforce more strong pessimism leveraging the prior on the random policy; the policy would not work as well as the worst $\beta$: $\hat{Q}^\beta(s,\mu(s)) \leq \min_{a \in B(s)} Q^\beta(s,a) = Y^\beta(s;\epsilon)$. Again, we use $\epsilon = 0.1$ as a safety margin, i.e. $\tau_2$ = 0.1. Certainly, the assumption that $\mu$ performs worse than the worst $\beta$ might not hold, but we found that it worked well empirically.
>
> ### How to set $\lambda$?
>
> Note that R is a function of state only, and the gradients coming from the term mostly affect the $\hat{Q}_\theta$ for off-policy $(s,a')$ input since $a' \sim \pi_\phi(s) \text{ or } \mu(s)$. The term enforces the pessimistic constraints, and $\lambda$ is a hyperparameter that controls the regularization power. Therefore, $\lambda$ is set to a specific value, and it does not change depending on $s$ or is not tuned over training iterations. We use $\lambda = 1 \text{ or } 0.1$ depending on the domain.
>
> ### Minor Comments
>
> * Difference among $\beta$, $\beta^*$, $\pi_\phi$, and $\tilde{\pi}$.
>
> $\beta$ is a behavior policy that is used to generate a dataset $D$, and it could be a mixture of different policies; for example, $\beta$ can be a mixture of a medium-performing policy and an expert policy.
>
> $\beta^*$ is the policy that selects the best action among actions executed by $\beta$; in the example above, it will be the expert policy.
>
> $\pi_\phi$ and $\tilde{\pi}$ is an approximation of $\beta^*$ inferred from $\hat{Q}_\theta$. $\pi_\phi$ is directly trained over $\hat{Q}_\theta$ as an actor-critic algorithm (See Algorithm 1 in L197-198 for the equation). $\tilde{\pi}$ is a nonparametric policy that explicitly restricts the search space of action to the set of actions given by the dataset $D$.
>
> * Definition of Support $B(s) = \text{supp} \beta(s)$.
>
> The support of the behavior policy is defined as follows: $B(s) = \text{supp} \beta(s) = \{ a \in \mathcal{A}: p_{\beta(s)}(a) > 0 \}$.
>
> ### References
>
> [1] Fujimoto, Scott, David Meger, and Doina Precup. "Off-policy deep reinforcement learning without exploration." International Conference on Machine Learning. PMLR, 2019.
> [2] Dabney, Will, et al. "Implicit quantile networks for distributional reinforcement learning." International conference on machine learning. PMLR, 2018.

---

> > ### Comment · Reviewer_q44u · 2021-09-01
> > **Updated review.**
> >
> > Hello Authors,
> >
> > I wanted to thank you for your time in updating the manuscript, I found it much more readable and hopefully others will benefit from this extra work.  I have updated my review to strong accept.

---

> ### Author Response · Authors · 2021-08-22
> **To Reviewer q44u [2/3]**
>
> ### State Value *Distribution* $Y$ and State Value Function $V(s)$; what is $\tau$?; Eq. (7) is unnecessary
>
> We apologize for the confusion. We missed introducing and defining the state value **distribution** $Y$. It seems like this is the source of the reviewer’s confusion. We will fix the manuscript.
>
> $Y$ is a mapping from a state $s$ to a distribution over return where its expectation is the value: $V(s) = \mathbb{E}[Y(s)]$. The value distribution is useful especially when we are dealing with stochastic environments or stochastic policies For example, assume that there exist two paths starting from the state $s_t$, where one is highly rewarding (+100) and the second is penalizing (-99). If a policy selects both paths with the same probability, then $V(s_t) = 0.5 * 100 + 0.5 * (-99) = 1$.  Also, assume that there is another state $s_{t'}$ which always provides the return of $0.5$, i.e. $V(s_{t'}) = 0.5$. Then, when it comes to selecting a better state based on $V$, it will always select $s_t$ over $s_{t'}$ since it has a higher value. However, in the domain where the risk of getting a negative return is extremely bad (e.g. a financial domain), it won't be an ideal solution. The state value distribution $Y$ can address such problems by considering a state value as a random variable.
>
> There are many different ways to describe a random variable; e.g. probability density function (PDF), cumulative distribution function (CDF), etc. Following Dabney et al. [2], we decided to use the quantile function to describe the random variable $V(s)$. The quantile function is the inverse of CDF, where it returns a minimum value $v$ from amongst all those values whose CDF value exceeds $\tau$: $\text{Prob}(V \leq v) = \tau$, or equivalently $Y(s;\tau) = \inf \{ v \in \mathbb{R}: \tau \leq \text{Prob} (V \leq v)\}$. Therefore, $Y(s;0.5)$ is median, $Y(s;\epsilon)$ and $Y(s;1-\epsilon)$ is (roughly speaking) minimum and maximum possible return at a given state.
>
> We admit that Eq. (7) is somewhat unrelated to our main algorithm since it is an empirical design choice we made to represent and to learn the state value distribution $Y(s)$. We will define $Y(s)$ more clearly instead of Eq. (7).

---

> ### Author Response · Authors · 2021-08-22
> **To Reviewer q44u [1/3]**
>
> We appreciate the insightful comments and advice to improve the paper. Here, we provide answers to the questions the reviewer raised, and we hope this can help clarify several issues. Please let us know if there is anything unclear. We will soon update the paper based on these comments, and we will let the reviewer know when it is ready.
>
> ### The motivation behind YOEO; Why does CEM or other policy optimization methods on top of unregularized $Q^\beta$ (trained w/ SARSA tuple) fail? (i.e. why is pessimism required even for $Q^\beta$?); How about training stochastic policy via BC to get the support of $\beta(s)$?; What is the main difference between Critic-Regularized Regression?
>
> We admit that the motivation behind YOEO is not fully introduced in Section 4; why does YOEO use pessimism even for $Q^\beta$? As the reviewer pointed out, the idea behind the pessimism can be easily justified with Figure A.1, which shows that unregularized $Q^\beta$ behaves more likely $V^\beta$ that makes CEM or other policy optimization fail. We will change the manuscript to include Figure A.1. at the beginning of Section 4, and we will add more explanations.
>
> It is a viable option to train a stochastic policy to approximate the behavior policy $\beta$ as done in BCQ [1]. If we first approximate $\beta$, we then can filter the good actions by querying $Q^\beta$ and clone the good actions only. This can be viewed as the first iteration of CRR (Note that CRR iterates the process by further training $Q^{\pi^{(k)}}$). We actually tried this approach, but the performance was poor in our early attempts. The problem we found is that the filtering is tricky when the dataset has very narrow action support. For example, assume that $\beta$ emits a specific action $a_t$ for a certain state $s_t$. Then, the advantage calculated with approximated behavior policy $\hat{\beta}$  using a CRR-type formula (i.e. $A(s_t,a_t) = Q^\beta(s_t,a_t) - \max_{j=1}^m Q^\beta(s_t,a^j), \text{ with  } a^j \sim \hat{\beta}(\cdot|s_t)$) can easily become negative or near-zero due to approximation error in $\beta$ and $Q^\beta$. Therefore, the only known, valid action $a_t$ for $s_t$ has a higher chance not to be used for cloning, and the policy can perform poorly. To this end, we moved our attention to implementing pessimism instead of directly approximating $\beta$.
>
> ### Why do we define $\hat{Q}^\beta$ with $\leq$ , not $\lt$ ?
>
> Thanks for pointing this out. The first step of the proof ($\arg\max_{a \in \mathcal{A}} \hat{Q}^\beta(s,a) = \arg\max_{a \in B(s)} \hat{Q}^\beta(s,a)$) can be technically false with $\leq$. For correctness, we will change the definition with $\lt$. Please note that the first and the second condition of the theorem implies $\hat{Q}^\beta (s,a') \leq \max_{a \in B(s)} Q^\beta(s,a) \ \forall s \in D, a' \in \mathcal{A}$, and we use this for the implementation.

---

### Official Review · Reviewer_LDRr · 2021-07-26

**Originality:** Good
**Technical Quality:** Fair
**Clarity Of Presentation:** Good
**Impact:** 4

**Recommendation:**

Weak Reject: I recommend rejecting the paper, but will not argue for my recommendation if the majority of other reviewers have a different opinion.

**Summary:**

This paper proposes a simple baseline for offline RL where the authors only perform the policy evaluation step once without performing policy optimization in an iterative loop like traditional actor-critic algorithms. The authors also conduct pessimistic regularization on the Q function of the behavior policy to ensure that it does not suffer from OOD actions. Empirically, the proposed method achieves competitive results and even best performances in certain cases in standard offline RL benchmarks despite being just a policy evaluation approach.

**Issues:**

1. Compare the latest CQL results.
2. Evaluate the method on more complex domains such as antmaze and kitchen.

**Reviewer Expertise:**

Very good: Comprehensive knowledge of the area

**Strengths And Weaknesses:**

The paper presents an interesting and strong baseline for offline RL, which is somewhat novel. The insight on the challenges of iterative off-policy actor-critic learning in offline RL is also intriguing and is overlooked by the community in the past. The authors also present strong empirical evidence that removing the iterative process and only keeping the policy evaluation step also yield strong performances in certain cases, which collaborating the claim that iterative actor-critic learning is hard in offline RL. The authors also provide a theoretical analysis of the proposed approach, which makes the paper more theoretically grounded.

However, I do have a few concerns over the paper, which I will discuss as follows.

First, despite that the results show that the method can achieve competitive results compared to prior offline RL methods, the domains where the authors evaluated on are either hard for all of the methods or generated by a single behavior policy except the medium-replay, medium-expert and random datasets in D4RL where the proposed method is underperforming. Therefore, I think the results are somewhat less surprising and a bit uninteresting since the proposed method (YOEO ($\pi^\tilde$)) can be viewed as a strong behavior cloning approach and should perform well in settings where data distribution is narrow and data quality is high. Besides, I believe the CQL results in the paper are not up-to-date. The CQL results in the NeurIPS camera-ready (https://proceedings.neurips.cc/paper/2020/file/0d2b2061826a5df3221116a5085a6052-Paper.pdf) are significantly better than the proposed method in almost all the domains. Therefore, I think the benefits of the method over SOTA offline RL methods are somewhat unclear.

Moreover, I think it would also be important to evaluate the method in settings such as antmaze and kitchen where the dataset is diverse and composed of undirected data. I think the proposed method could struggle there since simply learning the optimal behavior policy is not sufficient and also challenging in those settings.

============================

After reading the rebuttal, I still think that the YOEO does not perform well on wide datasets such as random and medium-replay, suggesting it could just be a strong BC method with some additional tricks. The variants of YOEO also doesn't perform well uniformly against CQL on all settings of kitchen. It is also unclear how the authors would pick the two variants of YOEO, which could just introduce additional hyperparameters for tuning. Therefore, I would keep my current score.

**Summary Of Recommendation:**

While the paper presents an interesting baseline for offline RL, given the two points on the insufficient empirical evidence mentioned in the Strengths And Weaknesses section, I would vote for a weak reject.

---

> ### Author Response · Authors · 2021-08-22
> **To Reviewer LDRr [2/2]**
>
> * Comparison to the latest CQL results
>
> Thanks for letting us know about the updated results. While most of the numbers remain the same, we were able to find updates on `hopper-medium` and `walker2d-medium`; the performance on `hopper-medium` increased by about 28%p, and the performance on `walker2d-medium` decreased by about 3%p. We will update the paper based on these new results. For reference, we also display the numbers in the table below. We also included the reproduced results done by Kostrikov et al. [1] who tried the code provided by the authors, as well as their own implementation. We only included the best between the two. Overall, it was possible to reproduce the reported numbers, but there are a few cases where Kstrikov et al., couldn't get the exact number, namely, `halfcheetah-random`, `hopper-medium-replay`, `walker2d-medium-replay`, and `hopper-medium`.
>
> The changes are not significant (especially when YOEO is compared against the best of the reproduced performance), and therefore, it does not affect the main argument of our paper; the performances of the current offline RL algorithms are similar to YOEO that only fits $Q^\beta$, and this is clear evidence that the algorithms do not fully exploit the potential advantage of the iterative process.
>
> |               |             | CQL (arXiv) | CQL (NeurIPS) | CQL (Best of Reprod.) | YOEO ($\pi_\phi$) | YOEO  ($\tilde{\pi}$) |
> | ------------- | ----------- | ----------- | ------------- | --------------------- | ----------------- | --------------------- |
> | random        | halfcheetah | 35.4        | 35.4          | 27.1                  | 4.5               | 5.7                   |
> |               | hopper      | 10.8        | 10.8          | 10.6                  | 12.2              | 12.3                  |
> |               | walker2d    | 7           | 7             | 10.0                  | 4.9               | 4.1                   |
> | medium-replay | halfcheetah | 46.2        | 46.2          | 44.9                  | 36.2              | 30.9                  |
> |               | hopper      | 48.6        | 48.6          | 31.6                  | 42.5              | 37.2                  |
> |               | walker2d    | 26.7        | 32.6          | 16.8                  | 41.9              | 30.1                  |
> | medium        | halfcheetah | 44.4        | 44.4          | 40.3                  | 45.1              | 43.6                  |
> |               | hopper      | 58          | 86.6          | 42.2                  | 71.9              | 62.5                  |
> |               | walker2d    | 79.2        | 74.5          | 77.3                  | 74.1              | 76.2                  |
> | medium-expert | halfcheetah | 62.4        | 62.4          | 58.6                  | 41.9              | 35.9                  |
> |               | hopper      | 111         | 111           | 111.3                 | 112.1             | 112                   |
> |               | walker2d    | 98.7        | 98.7          | 104                   | 89.5              | 84.3                  |
>
> * Evaluation on more complex domains, such as kitchen or antmaze
>
> We evaluated YOEO on the kitchen domain, and we were able to confirm a similar trend: YOEO achieves state-of-the-art results in the `kitchen-complete` dataset, and competitive results on the other two datasets, namely `-partial` and `-mixed`. We displayed the numbers in the table below. We are currently running the experiments on antmaze, and we will report the number in the comment as soon as it is ready.
>
> | Type    | Environemt | BC   | SAC- offline | BEAR | BRAC | AWR  | BCQ  | CQL ($\mathcal{H}$) | YOEO ($\pi_\phi$) | YOEO ($\tilde{\pi}$) |
> | ------- | ---------- | ---- | ------------ | ---- | ---- | ---- | ---- | ------------------- | ----------------- | -------------------- |
> | kitchen | complete   | 33.8 | 15           | 0    | 0    | 0    | 8.1  | 43.8                | 63.2              | 31.3                 |
> |         | partial    | 33.8 | 0            | 13.1 | 0    | 15.4 | 18.9 | 49.8                | 17.2              | 46.8                 |
> |         | mixed      | 47.5 | 2.5          | 47.2 | 0    | 10.6 | 8.1  | 51                  | 5.9               | 40.4                 |
>
> * Reference
>
> [1] Kostrikov, Ilya, et al. "Offline reinforcement learning with fisher divergence critic regularization." *International Conference on Machine Learning*. PMLR, 2021.

---

> ### Author Response · Authors · 2021-08-22
> **To Reviewer LDRr [1/2]**
>
> We sincerely appreciate the detailed review. The main concerns are (1) about the benefit compared to SOTA offline RL methods, and (2) the CQL results used for the comparison. Also, the reviewer asked for more experiments on more complex domains to confirm the empirical validity. Here, we clarify the main contribution and share the updated table with the latest CQL results. We are also conducting experiments for more difficult domains, so we hope that this addresses the reviewer’s concern.
>
> * Unsurprising results / The benefit compared to SOTA offline RL methods is unclear.
>
> We would like to emphasize that the goal of this research is to thoroughly examine whether existing iterative offline RL algorithms fully take advantage of inferring the optimal value function $Q^*$. YOEO ought to be outperformed by an algorithm that properly infers $Q^*$ since it only fits $Q^\beta$. From this point of view, whether the proposed method achieves state-of-the-art results or not is not essential. What matters is whether there is a large gap between YOEO and other methods. In short, we do not aim to propose a practical algorithm that everyone should use for offline RL. We are proposing a ***benchmark*** that all the sound offline RL algorithms should outperform.
>
> Our paper is important for the community since the experimental results call into question many current offline RL algorithms: YOEO showed competitive performance and even state-of-the-art performance in some domains (It still holds with the updated CQL results, please see the paragraph below). The results imply that the examined iterative offline RL algorithms failed to learn a valid $Q^*$and that there is much room for improvement. The community will benefit from such findings since they can guide researchers in the search for better algorithms.
>
> * YOEO as a strong behavior cloning approach / Only works well in settings where data distribution is narrow and data quality is high.
>
> It is a valid viewpoint to regard YOEO as a strong behavior cloning approach since the action space is limited to the support of the behavior policy implicitly (YOEO $\pi_\phi$) and explicitly (YOEO $\tilde{\pi}$). However, it is inaccurate to expect that YOEO can only work well when the given data distribution is narrow and the data quality is high. This is because we search for and execute the best action given a state. The empirical results also support our claim that YOEO can work well even when the data distribution is **not** narrow; YOEO shows competitive performance not only for `-medium` datasets but also for `-medium-replay` and `-medium-expert` datasets in which filtering the best action is required due to mixed data quality. Especially, the results on `walker2d-medium-expert` show the effectiveness of YOEO: BC without any action filtration failed catastrophically due to a bimodal action distribution while YOEO performed well.
>
> As we discussed above, YOEO selects an action based on $Q^\beta$ only, so YOEO may underperform compared to the policy optimized on top of $Q^*$. The results on `-random` datasets are such cases, and they indicate the iterative process is sometimes beneficial in offline RL. However, the competitive performance of YOEO in other datasets still calls into question the effectiveness of the current SOTA algorithms.

---

### Official Review · Reviewer_HJhz · 2021-07-26

**Originality:** Very Good
**Technical Quality:** Very Good
**Clarity Of Presentation:** Very Good
**Impact:** 4

**Recommendation:**

Strong Accept: I recommend accepting the paper and will argue for my recommendation even if other reviewers hold a different opinion.

**Summary:**

The paper introduces a simple baseline for Offline RL that attempts to recover the best action of the behavior policy. The authors fit state value function with Implicit Quantile Regression, while the state-action value function is trained using the median value of the distributional critic and a pessimistic constraint to get lower values for out-of-distribution actions.

**Issues:**

I would encourage the authors to discuss failure cases of this approach.

**Reviewer Expertise:**

Excellent: Expert knowledge on the topic of the paper

**Strengths And Weaknesses:**

### Strength
* Fitting value-critic using behavior data and then impossible pessimism only on Q is a simple to implement and test baseline for Offline RL. It avoids action extrapolation since it fits a value function using behavior policy and fits Q with a pessimistic constraint.
* The approach is interesting and theoretically sound.
* This simple baseline demonstrates strong empirical performance comparable to the state-of-the-art methods for Offline RL.

### Weaknesses
* I do not see any major weaknesses

**Summary Of Recommendation:**

I believe that the paper introduces an important baseline for Offline RL that will impact the design of algorithms and research benchmarks.

---

### Author Response · Authors · 2021-08-30
**Revised Paper Uploaded**

We updated the content of the manuscript representing the comments we provided for the reviewers; we included additional experimental results on the kitchen dataset, and we mainly edited the section 4 and 4.1 following the suggestions provided by the reviewer q44u. We hope the updates address the concerns of the reviewers.

---

### Meta-Review · Area_Chair_mCH9 · 2021-08-06

**Recommendation:** Accept (Poster)
**Confidence:** 4

**Metareview:**

The reviewers all like the conceptual idea of the algorithm presented in the paper. The authors' have provided substantial revision to placate Reviewer q44u's concerns. Reviewer LDRr still has some concerns about the baseline performing well only on some datasets and not others. I suggest for the final version, the authors add a paragraph or two discuss this. Indeed, I personally believe that one of the main reasons such a simple baseline performs so well may be because of inadequate benchmarks in the offline RL community.

---

### Decision · Program_Chairs · 2021-09-13

**Decision:**

Accept (Poster)

**Comment:**

The reviewers all like the conceptual idea of the algorithm presented in the paper. The authors' have provided substantial revision to placate Reviewer q44u's concerns. Reviewer LDRr still has some concerns about the baseline performing well only on some datasets and not others. I suggest for the final version, the authors add a paragraph or two discuss this. Indeed, I personally believe that one of the main reasons such a simple baseline performs so well may be because of inadequate benchmarks in the offline RL community.